# Microbiome of Ceca from Broiler Chicken Vaccinated or Not against Coccidiosis and Fed Berry Pomaces

**DOI:** 10.3390/microorganisms11051184

**Published:** 2023-04-30

**Authors:** Chongwu Yang, Quail Das, Muhammad A. Rehman, Xianhua Yin, Julie Shay, Martin Gauthier, Calvin Ho-Fung Lau, Kelly Ross, Moussa S. Diarra

**Affiliations:** 1Guelph Research and Development Center, Agriculture and Agri-Food Canada (AAFC), Guelph, ON N1G 5C9, Canada; 2Ottawa Laboratory (Carling) Research and Development, Canadian Food Inspection Agency, Ottawa, ON K1Y 4K7, Canada; 3Biological Informatics Centre of Excellence, AAFC, Saint-Hyacinthe, QC J2S 8E3, Canada; 4Summerland Research and Development Center, AAFC, Summerland, BC V0H 1Z0, Canada

**Keywords:** berry pomaces, broiler chickens, coccidiosis vaccine, microbiota, virulome, resistome, metabolic pathways

## Abstract

American cranberry (*Vaccinium macrocarpon*) and lowbush/wild blueberry (*V. angustifolium*) pomace are polyphenol-rich products having potentially beneficial effects in broiler chickens. This study investigated the cecal microbiome of broiler-vaccinated or non-vaccinated birds against coccidiosis. Birds in each of the two groups (vaccinated or non-vaccinated) were fed a basal non-supplemented diet (NC), a basal diet supplemented with bacitracin (BAC), American cranberry (CP), and lowbush blueberry (BP) pomace alone or in combination (CP + BP). At 21 days of age, cecal DNA samples were extracted and analyzed using both whole-metagenome shotgun sequencing and targeted-resistome sequencing approaches. Ceca from vaccinated birds showed a lower abundance of *Lactobacillus* and a higher abundance of *Escherichia coli* than non-vaccinated birds (*p* < 0.05). The highest and lowest abundance of *L. crispatus* and *E. coli*, respectively, were observed in birds fed CP, BP, and CP + BP compared to those from NC or BAC treatments (*p* < 0.05). Coccidiosis vaccination affected the abundance of virulence genes (VGs) related to adherence, flagella, iron utilization, and secretion system. Toxin-related genes were observed in vaccinated birds (*p* < 0.05) in general, with less prevalence in birds fed CP, BP, and CP + BP than NC and BAC (*p* < 0.05). More than 75 antimicrobial resistance genes (ARGs) detected by the shotgun metagenomics sequencing were impacted by vaccination. Ceca from birds fed CP, BP, and CP + BP showed the lowest (*p* < 0.05) abundances of ARGs related to multi-drug efflux pumps, modifying/hydrolyzing enzyme and target-mediated mutation, when compared to ceca from birds fed BAC. Targeted metagenomics showed that resistome from BP treatment was distant to other groups for antimicrobials, such as aminoglycosides (*p* < 0.05). Significant differences in the richness were observed between the vaccinated and non-vaccinated groups for aminoglycosides, β-lactams, lincosamides, and trimethoprim resistance genes (*p* < 0.05). Overall, this study demonstrated that dietary berry pomaces and coccidiosis vaccination significantly impacted cecal microbiota, virulome, resistome, and metabolic pathways in broiler chickens.

## 1. Introduction

Due to the high demand for chicken meat, intensive commercial poultry farming systems are becoming popular; however, their ability to minimize feeding costs and maximize production remains to be improved [1]. Further, when feeding the birds at high stock density in these systems, their growth performance can be easily compromised due to exposure to pathogenic bacteria such as *Clostridium perfringens*, *Escherichia coli*, and *Salmonella* serovars as well as stressors including heat, ammonia, and oxidative stresses [2,3,4]. Feed supplementation with sub-therapeutic concentrations of antibiotics can prevent infectious diseases and promote growth performance. However, this practice has been reported to promote antimicrobial resistance (AMR) and antimicrobial resistance genes (ARGs) in bacteria, which can be transmitted horizontally by mobile genetic elements from one species to another [5]. Antibiotic-resistant bacteria such as *E. coli* have been isolated from feces, carcasses, and the environment of antibiotic-free poultry production systems [6,7]. Despite this, there is an increasing demand for antibiotic-free poultry products worldwide [8]. Broilers raised without antibiotics (antibiotic-free) may face compromised growth performances and an increased risk of infectious diseases, such as necrotic enteritis (NE) [9]. The increasing pressures of disease in antibiotic-free chickens have led to the exploration of antimicrobial alternatives, including vaccines and phytogenic products.

Coccidiosis caused by *Eimeria* spp. is a common parasitic disease in broiler chicken farming. Coccidiosis is responsible for over $3 billion in economic losses annually in poultry production [10]. Live attenuated *Eimeria* vaccines have been used to control coccidiosis [11]; however, their efficacy could be affected by environmental conditions, and the application of these vaccines may induce inflammation which could compromise growth performance [12,13]. American cranberry (*Vaccinium macrocarpon*) and lowbush/wild blueberry (*V. angustifolium*) are important economic crops in North America. Using phenolic-rich berry pomace as a feed supplement has been reported to reduce gut lesions due to *C. perfringens* (NE) and *Eimeria* spp. (coccidiosis), to stimulate the growth of cecal beneficial bacteria, and to influence lipid and serum enzyme levels in broiler chickens [14,15]. Dietary cranberry and lowbush blueberry pomaces in feeds could also improve the quality and antioxidant capacity of chicken meat [16]. Limited studies have investigated the mechanisms behind the ability of these berry pomaces to modulate gut microbiome and host metabolism in broiler chickens.

Metagenomics Phylogenetic Analysis (MetaPhlAn) [17], Short Reading Sequencing Typing (SRST2) [18], AmrPlusPlus (AMR++) [19], and HUMAnN (v2.0) are tools frequently used to investigate microbiota taxonomy, virulome, resistome, and metabolic pathways, receptively from shotgun metagenomics sequencing data [20,21]. Monitoring the gut microbiome of berry pomace-fed chicken could help in understanding the mechanism of actions of these products to shape the gut microbiota and promote overall health. The main objective of this study was to analyze cecal microbiota, virulome, resistome, and metabolic pathways in American cranberry and lowbush blueberry pomace-fed broiler chickens vaccinated or not vaccinated against coccidiosis. 

## 2. Methodology

### 2.1. Berry Pomaces Preparation

The preparations and chemical compositions of the American cranberry (*Vaccinium macrocarpon*) (CP) and lowbush blueberry (*V. angustifolium*) (BP) pomaces used in this study were described in previous studies [15,22]. Briefly, a frozen hydraulic rack and cloth were used to press berry fruits at 1000 to 3000 psi for juice removal. The pomaces were then freeze-dried and ground through a 2 mm mesh screen using a cutting mill (SM 2000 Retsch, Haan, Germany). 

### 2.2. Chicken Trial Design and Sample Collections

The chicken feeding trial design, management, and sample collections have been described previously [15]. All protocols of this animal trail were approved (protocol #16-AV-314) by the Animal Care Committee of Center de Recherche en Sciences Animales de Deschambault (CRSAD) based on guidelines of the Canadian Council on Animal Care [15]. Briefly, 2700 male day-old Cobb 500 chicks were allocated randomly to 60 pens (45 birds/pen) separated into two groups with or without coccidiosis vaccines. The birds in both groups were assigned to five dietary treatments (pens/treatment), including basal diets (NC), basal diet with 55 ppm bacitracin as positive control (BAC), 1% cranberry (CP), 1% blueberry (BP) alone, or in combination (CP + BP). The diets were formulated to meet the nutritional recommendations for Cobb 500 [23]. At 21 day of age, the cecal contents from two birds/pen (4 pens/treatment) were collected (total of 8 birds/treatment). The collected cecal samples were transferred into sterile Whirl-Pak plastic bags (Nasco, Fort Atkinson, WI, USA) and stored at −20 °C before processing to generate a total of 40 pooled DNA samples for analysis. 

### 2.3. Genomic DNA Extraction and Metagenomics Data Analysis

Genomic DNA was extracted using the QIAamp DNA Stool Mini Kit (Qiagen, Hilden, Germany), as previously described [15]. Briefly, to increase the DNA quality and minimize biases to ensure reproductivity, bead-beating method and denaturants, including guanidine isothyocynate and β-mercaptoethanol, were used. Typically, FastQC (v0.11.8) and MultiQC (v1.6) were used to check the sequence quality of each sample and the dataset as a whole. Trimmomatic (v 0.38) was used to remove adapters, leading and trailing bases below a Phred score of 30. A sliding window was used to cut reads when the average base quality dropped below a Phred score of 15 and reads dropped less than 36 bases long after trimming. 

Whole-metagenome shotgun sequencing and quality control steps were performed at Genome Quebec Innovation Centre (McGill University, Montréal, QC, Canada). After TruSeq DNA libraries preparation, samples were processed by an Illumina HiSeq2000 platform, with four samples multiplexed per sequencing lane to generate 2 × 100 base paired-end (PE) reads from the 40 ceca DNA samples resulting in a total sequencing output of ca. 2.2 billion reads with an average of approximately 55 million reads per sample [15].

The composition of microbial communities (Bacteria, Archaea, Eukaryotes, and Viruses) from the shotgun metagenome sequence data from phyla to species levels was performed with MetaPhlAn2 (v 2.7.5) using the default parameters and the default MetaPhlAn database. Results were merged for multiple samples using the merge_metaphlan_tables script included with MetaPhlAn. The α-diversity (Chao, Shannon, and Simpson) indexes for both the richness and evenness of the sample and (Bray–Curtis’s dissimilarity) were computed with R’s vegan (v 2.5) package using MetaPhlAn’s taxonomic assignments. The association between microbial diversity and vaccination and/or treatment groups was tested by multivariate statistics and Analysis of Similarities (ANOSIM). Sample variation between groups or β-diversity was performed with Bray–Curtis’s dissimilarity as the distance measure between samples using the R package vegan v 2.5. For all the statistical analyses performed, a *p*-value < 0.05 was considered significant. 

The resistome analysis of the metagenomic datasets was performed with the AmrPlusPlus (AMR++) pipeline version 2.0.0 [19]. The virulence genes were analyzed with SRST2 v 0.2.0 [18] using the default parameters and the virulence factor databases (http://www.mgc.ac.cn/VFs, accessed 13 March 2019) formatted for SRST2 using CD-HIT version 4.7. Metagenome predictions were performed against the Kyoto Encyclopedia of Genes and Genomes (KEGG) database, and functional metabolic profiling of the metagenomic datasets was performed using HUMAnN2 (v 0.11.2) algorithm [20]. To apply statistical analysis, genes for virulence, antimicrobial resistance, and metabolic functions hit counts against hits to bacterial marker genes from MetaPhlAn were used for normalization.

### 2.4. Targeted-Resistome Sequencing

For the comprehensive evaluation of the sample resistome composition, a target-enrichment sequencing approach using a myBaits^®^ Custom DNA-Seq kit was also performed [24]. Unless otherwise specified, the hybridization-based capturing and enrichment of ARG target sequences present in the above-described dual-indexed metagenomic shotgun sequencing libraries were performed in accordance with the myBaits^®^ Custom kit’s manual v4.01, followed by Illumina paired-end sequencing. Briefly, pre-capture library amplification was performed using 100 ng of library DNA input, together with primers (reamp-P5/reamp-P7) [25] at 500 nM final concentration, and 1X KAPA high-fidelity polymerase (HiFi) HotSart ReadyMix (Roche, Laval, QC, Canada). The mixtures were heated for 2 min. at 98 °C, followed by nine cycles of 20 s at 98 °C, 30 s at 60 °C, and 45 s at 72 °C, before finishing with 5 min at 72 °C. The amplified libraries were purified using 1.8X volume of Agencourt AMPure XP beads (Beckman, Mississauga, ON, Canada) before being pooled into individual hybridization reactions consisting of amplified DNA (800 ng each) derived from two sample libraries. After 20 h of hybridization at 65 °C in the presence of the customized RNA probes provided in the myBaits^®^ kit, the probe-target complexes were allowed to bind to streptavidin-coated magnetic beads, and the unbound non-target DNA was later removed using the provided wash buffer. The resultant bead-bound, target-captured libraries were subjected to post-capture amplification that was performed using the exact conditions as described above, except 15 μL of the bead-bound library DNA was included as a template, and a total of 12 amplification cycles were used. Multiplex high-throughput sequencing of the target-enriched libraries was performed on an Illumina NextSeq 500 sequencer using the mid-output v2.5 kits (300-cycle) and a loading concentration of 1.5 pM with 1% PhiX spike-in to generate 2 × 150 bp paired-end sequences, with a targeted output of 10–20 million raw reads per sample.

### 2.5. Data Analysis

The effects of vaccination, dietary treatments, and their interactions on microbiota taxonomy, antimicrobial resistance, and virulence genes, as well as metabolic pathways data, were analyzed following a two (vaccinated or non-vaccinated) by five (dietary treatments) factorial design using the General Linear Mixed Model (GLMM) procedure of Statistical Analysis System version 9.4. Diets (treatments) and vaccination (yes or no) were used as sources of variation, and the individual pens as experimental units. The least significant difference (LSD) test was used to separate means whenever the *F*-value was significant. To test for statistically significant differences in the alpha-diversity data, the non-parametric Kruskal–Wallis one-way ANOVA test followed by Dunn’s post hoc test, with Bonferroni correction to adjust *p*-values for multiple pairwise comparisons, were performed using the functions “kruskal_test” and “dunn-test” from R package rstatix ver. 0.7.0. The beta-diversity was determined through principal coordinates analysis (PCoA) of Bray–Curtis’s dissimilarity performed through the “vegdist” and “pcoa” functions of R packages vegan and ape ver. 5.4-1, respectively. Significances of (dis)similarity between and within treatments were examined by permutational multivariate analysis of variance (PERMANOVA) and analysis of similarities (ANOSIM) using vegan functions “adonis” and “anosim” with 999 permutations, and by using R package pairwise Adonis ver. 0.4 function “pairwise.adonis” with Bonferroni correction and 999 permutations for pairwise comparisons between selected types. Spearman’s Rank correlation was applied to HUMAnN data, using the rcorr function from the Hmisc (v4.6) package in R, to analyze the correlation between microbial genera and metabolic pathways. A *p*-value of 0.05 was used to declare significance.

## 3. Results

### 3.1. Relative Abundance of Cecal Microbiota by MetaPhlAn

Based on the MetaPhlAn analysis, the relative abundance of cecal microbiota of 21-day-old broiler chickens at the kingdom, phylum, family, genus, and species levels was affected by vaccination and dietary treatments, as shown in Table 1.

Kingdom. Bacteria and viruses were more abundant (*p* < 0.05) in birds fed CP, BP, and CP + BP than in those fed NC or BAC, regardless of vaccination. Vaccinated birds tended to have a high relative abundance of Eukaryotes when compared to non-vaccinated birds. 

Phylum. The most dominant bacterial phylum was *Firmicutes*, representing over 70% with the lowest relative abundance being found in vaccinated birds *(p* < 0.01). In the vaccinated group, feeding with CP, BP, or CP + BP increased (*p* < 0.05) the abundance of *Firmicutes* compared to birds fed NC or BAC. *Proteobacteria* were more abundant in the vaccinated group than in the non-vaccinated group (*p* < 0.05). 

Family. The highest (*p* < 0.05) abundance of *Lactobacillaceae* was observed in the non-vaccinated birds, while the vaccinated group showed the highest (*p* < 0.01) abundance of *Enterobacteriaceae*. A higher abundance of *Lactobacillaceae* along with a lower abundance of *Retroviridae* were observed in birds fed CP, BP, or CP + BP than in those fed NC or BAC in the vaccinated or non-vaccinated group (*p* < 0.05). 

Genus. The predominant genus was *Lactobacillus* (50%), which was more abundant in the non-vaccinated group than in the vaccinated group of birds fed CP, BP, or CP + BP compared to NC or BAC. *Escherichia* genus was more abundant in the vaccinated group, with the lowest relative abundance being observed in all berry pomace treatments compared to NC and BAC (*p* < 0.05)

Species. Of the 14 bacterial species identified, the relative abundances of *Lactobacillus crispatus*, *E. coli*, *unclassified Escherichia*, and *Avian Endogenous retrovirus* were affected (*p* < 0.05) by vaccination and/or dietary treatments. In general, birds from the vaccinated group showed a lower (*p* < 0.05) abundance of *L. crispatus* when compared to those in the non-vaccinated group. Birds fed CP, BP, or CP + BP in the vaccinated or non-vaccinated group showed the highest (*p* < 0.05) abundance of *L. crispatus* compared to NC or BAC treatments. A reduced relative abundance of *Escherichia* spp. was observed in CP-, BP- or CP + BP-fed birds from the vaccinated group, while in the non-vaccinated group, only birds fed BP- and CP + BP-fed birds showed a decreased (*p* < 0.05) abundance of *Escherichia* spp. *Eimeria tenella* (known to cause lesions on ceca), which was the only *Eimeria* spp. detected in the vaccinated group, while no *E. tenella* was found in the non-vaccinated group, showing the validity of our experimental model. Avian *endogenous retrovirus* was more abundant in vaccinated birds than in non-vaccinated birds (*p* < 0.05). 

Alpha-diversity and beta-diversity. For taxa richness and evenness, there were no significant differences between the vaccination groups or dietary treatments as indicated by α-diversity indices (Chao, Shannon, and Simpson). The PCoA plot using PERMANOVA for Unifrac weighted β-diversity showed that the cecal microbiota of the vaccinated and non-vaccinated birds clustered separately (*p* < 0.05).

### 3.2. Virulence Genes (VGs) by SRST2 

The overall relative abundance of bacterial VGs in 21-day-old broiler chicken ceca was modulated by bacitracin and berry pomaces feeding regardless of coccidiosis vaccination (Figure 1 and Figure 2). Descriptions of the trend for individual VG factor categories are presented below.

Adherence. The highest abundance of genes related to adherence was observed in vaccinated birds (*p* < 0.05). About 14 of VGs (*aatA*, *agn43*, *cps*, *ecp*, *eha*, *gnd*, *kps*, *ompD*, *pix*, *sfpC*, *upaG_ehaG*, *z2200*, and *z2206*) were more prevalent in vaccinated birds when compared to non-vaccinated ones (Figure 2). However, the *aci* and *smu* genes were more prevalent in birds fed BP, CP, or CP + BP than those in NC and BAC groups (*p* < 0.05). Significant interactions between vaccination and dietary treatments were observed for the prevalence of *aatA*, *csg*, *rml*, and *smu* genes.

Flagella. As shown in Figure 2, several genes related to flagella, including *flg*, *flk*, and *ppdD,* were more prevalent in vaccinated birds than in no-vaccinated birds (*p* < 0.05). However, the *flk* gene was less prevalent in berry pomace-fed birds (*p* < 0.05). Interactions between vaccination and dietary treatments were observed for the *fli* and *ppdD* gene prevalence (*p* < 0.05). 

Iron utilization. About six iron-uptake and transport-related genes (*fes*, *fur*, *hma*, *iutA*, *sit*, and *vat*) were more prevalent (*p* < 0.05) in vaccinated birds. No dietary treatment effects were observed. 

Secretion systems. The *APECO1*, *c33*, *ECABU*, *EcE24377A*, *ecs*, *esp*, *etrA*, *iagB*, *UMNK88_238*, and *UTI189* genes were more prevalent in vaccinated birds than non-vaccinated birds. Berry feeding tended to decrease the prevalence and abundance of *iagB* and *vgrG*. 

Toxin. Vaccinated birds showed the highest abundance of toxin-related genes such as *astA*, *cba*, *cdi*, *EcSMS35_B0007*, *pECS88_0104,* and *SPA1306* (*p* < 0.05). In the vaccinated group, dietary CP and CP + BP tended to decrease the prevalence of *col* compared to birds fed BAC (0.05 < *p* < 0.1). 

Miscellaneous. Vaccinated birds showed the highest abundance of miscellaneous genes (*p* < 0.01). The affected genes included *c33*, *ecs*, *esp*, *etrA*, *iagB*, and those associated with the APECO1 plasmid and *E. coli* UMNK88_238 and UTI189 isolates. Birds fed CP, BP, or CP + BP tended to increase the prevalence of *plr_gapA* and *tig_ropA* compared to birds fed NC or BAC. 

### 3.3. Antimicrobial Resistant Genes from Shotgun Metagenomics

Overall, shotgun metagenomics analysis using AMR++ detected in the ceca of 21-day-old broiler chickens about 560 ARGs (≥10 hits), conferring resistance to aminocoumarins, aminoglycosides, bacitracin, cationic antimicrobial peptides, elfamycins, fluoroquinolones, fosfomycin, fusidic acid, glycopeptides, lipopeptides, MLS, metronidazole, phenicol, rifampin, sulfonamides, tetracyclines, thiostrepton, trimethoprim, tunicamycin, β-lactams, and multiple drugs (efflux pumps). According to ANOSIM and PERMANOVA, no dietary treatment or vaccination effects were observed for the distribution of ARG classes (*p* > 0.05). However, Kruskal–Wallis tests per class of ARGs showed vaccination effects for genes conferring bacitracin (*p* = 0.02), trimethoprim (*p* = 0.04), cationic antimicrobial peptides (*p* = 0.001), and multi-drug (*p* = 0.01) resistance (Figure 3). The highest abundances of the last two ARGs classes were found in vaccinated and bacitracin-fed birds, while genes conferring resistance to bacitracin were more abundant in both bacitracin-fed and blueberry pomace-fed vaccinated birds (Figure 3). Among the individual ARGs detected, more than seventy-five were significantly affected by dietary treatments and/or vaccination (Figure 4).

Resistance-nodulation-division (RND) family transporters. The RND efflux pumps in Gram-negative bacteria are in the cytoplasmic membrane to actively transport substrates. About 22 RND multi-drug effluxrelated ARGs were significantly affected by vaccination and/or dietary treatment (*p* < 0.05). In general, the highest abundance of these ARGs was found in vaccinated birds (*p* < 0.05). Dietary CP, BP, and CP + BP in the vaccinated group showed the lowest abundance of RND-ARGs, including *acrBDEFS*, *asmA*, *baeR*, *cpxAR*, *crp*, *marA*, *mdtABCDE*, *gadX*, and *mexE* (*p* < 0.05). Interactions between vaccination and dietary treatments were observed for relative abundances of some of the ARGs, including *acrBDEFS*, *baeR*, *cpxA*, *cpxAR*, *crp*, *marA*, *mdtBCDE*, *gadX*, and *mexET.*

Major facilitator superfamily (MFS). The MFS encompasses thousands of active and passive transporters, including multi-drug efflux pumps. About 21 ARGs related to MFS multi-drug efflux pumps, including *mdtGHLNOP*, *mdfA*, *emrBDKRY*, *pmrABCF*, *cmrA,* and *tetHJ* were affected by vaccination and/or dietary treatments (*p* < 0.05). Birds fed CP, BP, or CP + BP showed the lowest (*p* < 0.05) abundance of nine ARGs, including *mdtHLNOP*, *mdfA*, *rosA*, and *pmrAB*.

Other efflux pump-related ARGs. Efflux pump genes including ATP-binding cassette transporters (ABC-transporters: *phoPQ* and *macB*), multi-drug and toxic compound extrusion (MATE: *mdtK*), small multi-drug resistance (SMR: *emrE*), MFS/RND (*evgAS* and *hns*), and RND/MFS/ABC (*soxS* and *tolC*) were affected by vaccination and/or dietary treatments (*p* < 0.05). Birds receiving vaccines showed the highest abundance of *phoPQ*, *mdtK*, *emrE*, *evgAS*, *hns*, *soxS*, and *tolC* (*p* < 0.05). In vaccinated birds, the lowest abundances of *emrE*, *evgA,* and *tolC* were found in CP, BP, and CP + BP treatments. Vaccination and dietary treatments showed interactions for eight genes (*mdtK*, *emrE*, *phoQ*, *evgAS*, *soxS*, *tolC*, and *macB*).

Target-mediated mutation, Modifying/hydrolyzing Enzymes. The abundances of two target-mediated mutation genes (*parE*, *gyrB*) were affected by vaccination and/or dietary treatment, while the *gidB*, *rmtA*, *pare,* and *gyrB*, as well as *mecA* and *pbp4B* genes, were more abundant (*p* < 0.05) in vaccinated birds than in non-vaccinated birds. 

More than twenty genes conferring resistance to aminoglycosides were detected, with the most abundant being *ant6*, *aac6-prime*, *aph2_dprime*, *aac3*, *aph3_prime*, and *aph6*. Genes including *ant3_dprime*, *aph2_dprime*, *aph4*, and *aph6* were significantly affected by vaccination and/or dietary treatments. Several β-lactam resistance genes, including *bla*_OXA_, *bla*_SHV_, *bla*_TEM_, *bla*_AMPH_, *bla*Z, *bla*_CEPA_, *bla*_CMY_, *bla*_CTX_, *bla*_Fox_, *bla*_GES_, *bla*_IMI_, and *bla*_KPC_ were detected with the most abundant being *OXA* followed by *CTX*, *CMY*, *SHV*, and *ampH*. The *ampH*, *CTX*, *GES*, and the class A β-lactamase *HER* were significantly affected by vaccination and/or dietary treatments (Figure 4A). Interactions between vaccination and dietary treatments were observed on the *ampH*, *cepS,* and *CTX.* Other β-lactam resistance genes, including *mecA* (conferring resistance to methicillin and penicillin-like antibiotics) and its associated *mecBCI* genes, were identified regardless of vaccination and diets.

Macrolides, lincosamides, streptogramins (MLS), and glycopeptides. The MLS (*erm32*, *ermN*, and *vatE*) and glycopeptides (*vanHD*, *vanSN*, *vanSO*, *vanTE*, *vanHM*, *vanSF*, and *vanXB*) resistance genes were affected by vaccination and/or dietary treatments. Vaccinated birds showed the highest (*p* < 0.05) abundances of *ompF*, *omp36*, and the *vanBDFHMSX* operon and the lowest abundance of *vatE* (Figure 4A). Birds fed CP, BP, or CP + BP showed a reduced (*p* < 0.05) abundance of *ompF*, *erm32*, *vanTE*, and *vanSF*. 

Other antimicrobials. About thirty-four genes conferring resistance to tetracycline have been found in studied ceca regardless of dietary treatment and vaccination, with the most abundant being *tetW* (found in a wide range of anaerobic intestinal bacteria including *Lactobacillus* and *Bifidobacterium*) followed by *tet32*, *tetO*, *tet44*, *tet40*, *tetM*, *tetL*, *tetA*, and *tetR*. Genes conferring resistance to other antimicrobials, including lincosamides (*lnuA*), lipopeptides (*cls*), bacitracin (*bacA*), peptides (*arnA*), fosfomycin (*fosB*), phenicol (*catB*), pyrazinamide (*pncA*), sulfonamide (*sul1*), thiostrepton (*tsnR*), and trimethoprim (*dfrG*, *dhfR*) were significantly affected by vaccination and/or dietary treatments (*p* < 0.05). Vaccinated birds showed the highest abundance of *cls*, *bacA*, *catB*, *pncA*, *sul1*, *tsnR*, and *dfrG*) but the lowest abundance of *lnuA* and *fosB* (*p* < 0.05). Dietary CP, BP, and CP + BP decreased the abundance of *bacA* and *arnA* compared to birds fed BAC. Interactions between vaccination and dietary treatments were noted for the abundance of *bacA*, *arnA,* and *dhfR* (Figure 4A).

### 3.4. Antimicrobial Resistance Genes by Targeted-Resistome Sequencing

A total of 130 ARGs were detected using a bait capture, as shown in Figure 4B. The most commonly detected t ARGs included those conferring resistance to aminoglycosides (*n* = 40), MLS (*n* = 28), tetracyclines (*n* = 17), and β-lactams (*n* = 10). In terms of the overall resistome structure, a significant difference in the alpha-diversity index of Shannon was observed between dietary treatments (*p* = 0.037) according to the Kruskal–Walli’s statistics, with BP treatment being distant from other groups. For individual antibiotic classes, the alpha diversity index of Shannon (*p* = 0.025) and Simson (*p* = 0.027) both showed a difference between birds fed blueberry pomace and other treatments for aminoglycoside resistance genes. Furthermore, significant differences in the richness were observed between the vaccinated and non-vaccinated groups for aminoglycoside (*p* = 0.011), β-lactam (*p* = 0.047), lincosamides-resistant (*p* = 0.034), and trimethoprim (*p* = 0.011) resistance genes, which is in alignment with the results from AMR++.

The β-diversity for the overall resistome and aminoglycoside resistance genes are presented in Figure 5A,B. Based on both weighted and un-weighted principal coordinate analysis (PCoA, Figure 5A), significant resistome dissimilarity was observed between vaccination groups (weighted *p* = 0.012; unweighted *p* = 0.009) with the Adonis-statistics suggesting that vaccination status only explained a small proportion of the observed dissimilarity (weighted R2 = 0.06; unweighted R2 = 0.09). The difference in overall resistome diversity was also observed among treatments based on weighted (*p* = 0.017), but not unweighted (*p* = 0.132), Bray–Curtis’s distance. For aminoglycoside resistance genes (Figure 5B), no difference was detected among treatments (*p* = 0.122), but significant dissimilarity was detected between vaccination groups (weighted *p* = 0.011; unweighted *p* = 0.008).

### 3.5. Correlations between Microbial Genera and ARGs

Correlations between microbial taxa at the genus level and antimicrobial resistance classes are shown in Figure 6. In general, microbial genera showed different correlations with ARGs. The genus *Escherichia* correlated with the genes including β-lactam, cationic antimicrobial peptide, fluoroquinolone, and multi-drug resistance genes. Glycopeptides, lipopeptides, and phenicol resistance genes were associated with *Enterococcus*, *Staphylococcus*, and *Vibrio*, respectively. Aminocoumarin, aminoglycoside, rifampin, and sulfonamide resistance genes correlated with *Shigella*, *Enterococcus*, *Streptococcus*, and *Escherichia*, respectively, while tetracycline resistance genes were mainly associated with *Bifidobacterium*, *Butyrivibrio*, *Campylobacter*, *Escherichia*, and *Streptococcus.* The MLS resistance genes correlated with *Clostridium*, *Enterococcus*, *Lysinibacillus*, *Salmonella*, and *Streptococcus*.

### 3.6. Correlations between Virulence Categories and ARGs

Correlations between virulence categories and ARGs are shown in Figure 7. As shown in Figure 7A, the abundance of lipopeptide resistance genes was negatively correlated with flagella-related genes. The cationic antimicrobial peptide, multi-drug, and sulfonamide resistance genes positively correlated with virulence gene categories, including adherence, flagella, and iron utilization. In non-vaccinated birds, only genes conferring cationic antimicrobial peptide resistance correlated positively with adherence, flagella, and iron utilization genes, while only metronidazole resistance genes were associated with toxin genes (Figure 7B).

Dietary treatments. In NC-treated birds, only the tunicamycin resistance genes showed a negative correlation with virulence genes of the secretion systems category (Figure 7C), while eight (aminoglycoside, bacitracin, cationic antimicrobial peptide, fluoroquinolone, multi-drug resistance, sulfonamide, trimethoprim, and β-lactam), resistance genes showed positive correlations with several virulence genes categories in bacitracin treatment (Figure 7D). In CP-fed birds, aminocoumarin, aminoglycoside, elfamycin, fluoroquinolone, fosfomycin, lipopeptide, MLS, rifampin, and tetracycline resistance genes showed negative correlations with several virulence gene categories (Figure 7E). In BP-fed birds, the cationic antimicrobial peptides showed positive correlations with flagella, iron utilization, and secretion system genes, while the trimethoprim resistance genes correlated with the secretion system genes (Figure 7F). Except for toxin genes, no correlations were observed between metronidazole resistance genes and virulence genes categories in the CP + PB-fed group (Figure 7G). In this group, no correlations were found between phenicol or trimethoprim resistance genes and virulence except for flagella genes (Figure 7G). 

### 3.7. Metabolic Pathways

The effects of vaccination and dietary treatments on the abundance of metabolic pathways of the ceca microbiome are shown in Figure 8. Of the 37 pathways, 14 were significantly (*p* < 0.05) or tended (0.05 < *p* < 0.1) to be affected by vaccination and/or dietary treatments. Two major groups were observed. The group 1 (G1) consisted of vaccinated birds fed BP, CP, and CP + BP, as well as non-vaccinated birds fed NC and the grope 2 (G2) consisted of vaccinated birds fed NC and BAC, as well as non-vaccinated birds fed BAC, BP, CP, and CP + BP.

The abundances of genes associated with aromatic compounds degradation, fatty acid and lipid degradation, glycolysis variants, metabolic regulators, non-carbon nutrients, TCA variants, and other energy were lower (*p* < 0.05) in ceca of vaccinated birds when compared to non-vaccinated birds. The tendency (0.05 < *p* < 0.1) of the increased number of genes related to aromatic compounds biosynthesis, electron transfer, lipid biosynthesis, and photosynthesis was observed in vaccinated birds compared to non-vaccinated birds. Furthermore, birds fed CP, BP, or CP + BP tended (0.05 < *p* < 0.1) to show a higher abundance of genes of nucleic acid processing than those fed NC or BAC.

### 3.8. Correlations between Bacterial Genera and Metabolic Pathways

The correlations between bacterial genera and metabolic pathways in vaccinated and non-vaccinated birds are shown in Figure 9. In the vaccinated group, eight bacterial genera (*Bacteroides*, *Blautia*, *Clostridium*, *Erysipelotrichaceae*, *Faecalibacterium*, *Flavobacterium*, *Lactobacillus*, and *Subdoligranulum*) were negatively correlated with at least one of eighteen identified metabolic pathways genes (Figure 9A). *Clostridium* was negatively correlated with the metabolic regulators, while *Lactobacillus* showed negative correlations with ten metabolic pathway genes, including amino acid degradation, aromatic compounds biosynthesis and degradation, fatty acids and lipids degradation, metabolic regulators, glycolysis variants, non-carbon nutrients, nucleic acid processing, polyamine biosynthesis, and TCA variants. *Escherichia* (*n* = 21), *Klebsiella* (*n* = 18), and *Proteus* (*n* = 15) showed positive correlations with most metabolic pathways, while *Lactobacillus* showed positive correlations with four metabolic pathways. *Enterococcus* positively correlated with amino acid biosynthesis, carbohydrate biosynthesis/degradation, and nucleic acid processing pathways (Figure 9A). In the non-vaccinated group, *Burkholderia* was the only bacterial genus that showed negative correlations with amino acid biosynthesis and carbohydrate biosynthesis/degradations (Figure 9B). Positive correlations were observed between *Escherichia* and eight metabolic pathways, while *Lactobacillus* showed positive correlations with 12 of the identified metabolic pathways.

## 4. Discussion

Coccidiosis vaccines have been widely used as an efficient way to prevent the incidence of coccidiosis and, indirectly, necrotic enteritis, but they may result in compromised growth performance [26]. Cranberry and blueberry pomaces have been shown to alter the cecal microbiota and blood metabolites of broiler chickens in birds receiving coccidiosis vaccines [15]. It has been reported that ceca harbor more diverse and stable microbial communities at day 21 than other intestines, such as the ileum [26]. Thus in this study, shotgun metagenomics analysis was applied to evaluate the impacts of dietary blueberry and cranberry pomaces on cecal microbiota, resistome, virulome, and metabolic pathways in 21-days-old broiler chickens.

MetaPhlAn is frequently used to enhance metagenomics taxonomic profiling using clade-specific marker genes, allowing identifying >17,000 reference genomes (>13,500 bacteria and archaea, >3500 viruses, and >110 eukaryotes) [27]. MetaPhlAn was used to determine microbiota taxonomic compositions in this study rather than Kraken in our previous study [15]. The chicken ceca host the largest number of microbes, although the composition changes are based on age, feed ingredients, and housing conditions [28]. In general, significant effects of coccidiosis vaccination were observed on the microbiota. Avian endogenous retrovirus, which has been associated with retarded muscle growth rate, reduced egg production, and compromised growth performance in poultry [29], was the only virus detected in studied ceca and found to be affected by treatments. In vaccinated birds, *E. tenella* was the only identified *Eimeria* spp. that specifically infects cecal epithelial cells available in the live coccidiosis vaccine [30]. The administration of coccidiosis vaccination could be the possible reason for the high *E. tenella* abundance in vaccinated birds.

Abnormal gut microbiotas, characterized by depletion of commensal bacteria, such as members belonging to the *Firmicutes* and *Bacteroidetes* phyla, and enrichment of *Proteobacteria* and the *Bacillus* subgroup of *Firmicutes*, has been associated with Crohn’s disease and ulcerative colitis [31]. In the present study, a clear decrease in *Firmicutes,* along with an increase in *Proteobacteria,* were observed in vaccinated birds. This observation could also reflect the impacts on the VGs, and AMR contents of the studied broiler ceca, as described below. Berry pomace feeding consistently increased the relative abundance of *Lactobacillus,* with *L. crispatus* representing over half of cecal *Lactobacillus* in the non-vaccinated group. In contrast, in-feed BAC was associated with *Escherichia* spp. Despite the fact that most *E. coli* strains are harmless commensals, pathogenic strains such as avian pathogenic *E. coli* can cause severe respiratory and systemic infections (colibacillosis) in birds [32,33]. Cranberry pomace extracts have been reported to decrease motility and biofilm formation of pathogenic *E. coli* [34,35]. The berry pomace as a feed ingredient may provide undigested nutrients (fibers) for fermentation to produce compounds that may stimulate the growth of beneficial bacteria, including some *Lactobacillus* species found in the chicken gut because of their ability to de-conjugate bile acids and acid tolerance ability [28]. Beneficial gut bacteria are known to produce vitamins and may competitively inhibit pathogenic ones by producing bacteriocins, and organic acids. These organic acids can be used by bacteria in anaerobic conditions to produce butyrate for energy formation and anti-inflammatory activities [36].

The broiler gut microbiome is associated with gut health, and vaccines whose quality could depend on various parameters and environmental factors against *Eimeria* have been used to protect against coccidiosis [13]. In broilers, coccidiosis vaccination seems to affect bird’s performance due to cell-mediated immune responses, including pro-inflammatory cytokine and mucin production, to the oocyst in vaccines [26,37]. In the present study, an increased abundance and prevalence of bacterial VGs, in parallel with the elevated *Proteobacteria* population, were observed in vaccinated birds. The origin and bacterial host of these VGs remain to be established. Mucins are highly glycosylated glycoproteins constituting the main component of the mucus secreted by goblet cells of the epithelial cells [38]. Since relationships may exist between gut bacteria and mucin composition [39,40], the high prevalence of VG observed in the present study could be related to factors including mucin production in vaccinated birds. Despite the unknown reasons for the increased prevalence of VGs after receiving a live oocyte vaccine, our results provided new insights into the potential impacts of live oocyst vaccines on bacterial virulence genes in the broiler gut. Further studies are warranted to reveal the mechanisms. The lower prevalence of some VGs in birds fed berry pomaces may be due to the anti-virulence effects of some of their compounds, such as proanthocyanins [41,42]. Down-expression of genes involved in adhesion, motility, and biofilm formation, and upregulation of genes involved in iron metabolism and stress responses in *Salmonella* and *E. coli* from broiler have been reported [34,43].

Antimicrobial resistance is a well-known economic and public health issue. Farm managements as well as environmental and genetic factors could contribute to the prevalence and spread of AMR and their related ARGs. Both the shotgun metagenomics analysis and targeted-resistome sequencing, which is more accurate and sensitive [44], methods were applied to investigate cecal ARG contents. In the present study, shotgun metagenomics analysis showed that ARGs were more abundant and prevalent along with the enrichment of *Proteobacteria* in vaccinated birds than in non-vaccinated birds. The relationship between *Proteobacteria* and increased ARGs needs to be established as several members of other bacteria phyla correlated with various ARGs. A low abundance of ARGs was observed in berry pomace-fed birds than in those fed basal and bacitracin diets. The ARGs that showed a decreased prevalence and abundance in berry pomace-fed birds include those related to efflux pumps and access prevention (permeability reduction). Phenolic compounds in berries affected bacterial cell structure, cell membrane synthesis, microbial adhesions, and biofilm formations [45,46,47]. Proanthocyanidins in cranberry pomace have been reported to interfere with intrinsic antibiotic resistance mechanisms in bacteria [48]. The present study revealed for the first time that AMR and its related genes in ceca could be promoted by coccidiosis vaccination; however, berry by-products such as pomaces in feed could alleviate this. However, results from the more-sensitive targeted resistome sequencing approach seemed to be consistent with previous studies reporting that the coccidiosis vaccine can ameliorate drug resistance in poultry [2,11]. As for the virulence genes, further studies are required to understand the mechanisms of the impacts of coccidiosis vaccines on AMR.

High abundances of metabolic pathways related to nutrient utilizations, such as aromatic compound degradation, were observed in the vaccinated group when compared to the non-vaccinated group. This indicated that the coccidiosis vaccine could enhance host nutrient utilization, which may explain the positive impacts of coccidiosis vaccination on energy and lipid utilization [49,50]. Berry pomaces have been reported to change the abundances of cecal microbiota and blood metabolites [15]. This may explain, in part, different correlations observed between microbiota and metabolic pathways in berry pomace-supplemented diets compared to basal and bacitracin diets; however, the mechanisms are currently unknown. In addition, a strong relationship between the nutrient metabolism of cecal microbiota and the nutrient utilization of the host (birds) has also been demonstrated in previous studies [51,52]. *Escherichia* was negatively correlated with the nutrient utilization in birds fed cranberry pomace, which is known to be rich in polysaccharides. It is noteworthy that the metabolism of polysaccharides has been previously reported to be negatively correlated with *E. coli* [53]. The amino acid degradation and fatty acid and lipid degradation pathways were positively correlated with *Escherichia* in this study in all treatments. A previous study reported that co-infection with *E. coli* could upregulate the metabolic pathways of linoleic acid, taurine, and arachidonic acid [54]. The positive correlations between *Lactobacillus* and carbohydrate metabolism in birds fed cranberry and blueberry pomace may indicate the important roles of berry pomace in the energy utilization of the host. Moreover, improved growth performance in broiler chickens by increasing carbohydrate metabolism after supplementations of *L. acidophilus* was also reported previously [55]. The present study reported that virulence and AMR were correlated; no strong evidence has ever been reported about such correlations using metagenome analysis of broiler ceca [56].

To contribute to identifying issues and opportunities in developing efficient and safe production practices, major poultry production systems and their impacts on AMR have been reviewed [57]. It has been reported that different diets could contribute to the transmission of ARGs by remodeling the intestinal environment, which could affect exogenous ARGs to change the intestinal resistome [58]. The present study also showed that dietary treatments and vaccination could impact the relationship between virulence categories and AMR classes which paves the way to understanding the relationship between AMR and pathogenicity.

In conclusion, the metagenomics analysis of ceca indicated that different diets (American cranberry and wild blueberry pomaces) under different management practices (coccidiosis vaccination) can significantly alter the structure of the microbial community and microbiome which could impact the physiologic and health status of male broiler Cobb 500. Similar future works on other broiler bird types and sex are warranted.

## Figures and Tables

**Figure 1 microorganisms-11-01184-f001:**
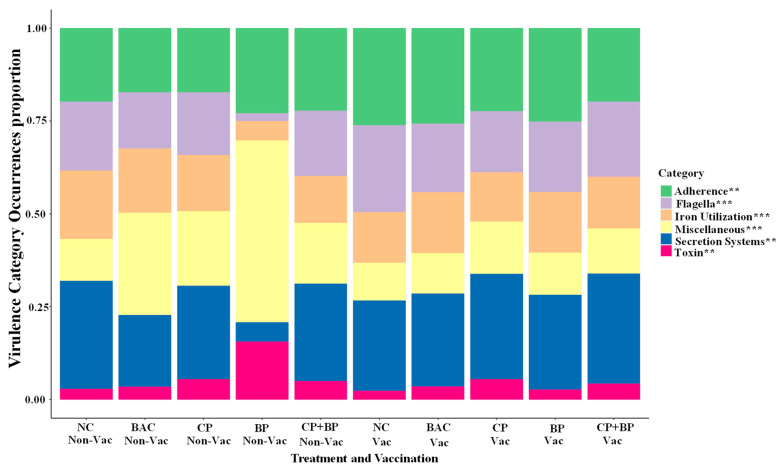
Relative abundances of the virulence genes categories detected by SRST2. Non-Vac, non-vaccinated group; Vac, vaccinated group; Con (NC), basal diet; Baci, 55 ppm bacitracin; CP, 1% cranberry; BP, 1% blueberry; and CP + BP, a combination of 1% cranberry and blueberry. Significantly affected by dietary treatment and/or vaccination: **, *p* < 0.05 and ***, *p* < 0.01.

**Figure 2 microorganisms-11-01184-f002:**
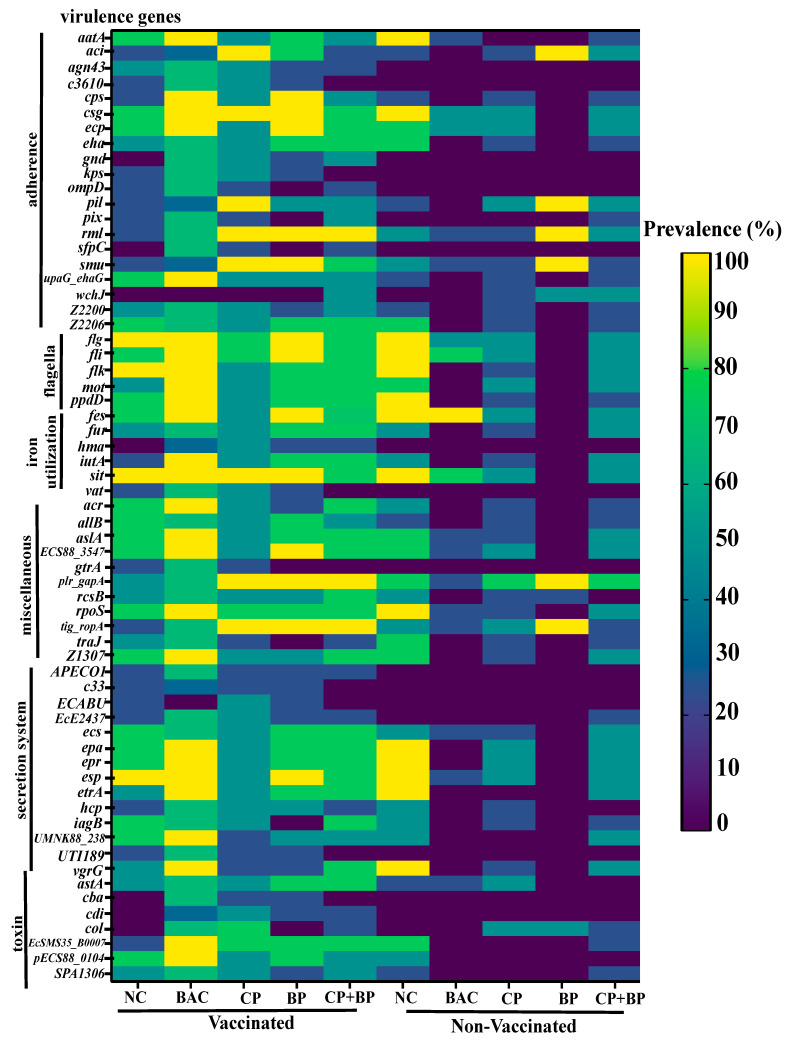
Heatmap showing changes in the prevalence of 63 VGs in studied dietary treatment groups and/or vaccination. NC, basal diet as negative control; BAC, 55 ppm bacitracin; CP, 1% cranberry; BP, 1% blueberry; and CP + BP, a combination of 1% cranberry and 1% blueberry.

**Figure 3 microorganisms-11-01184-f003:**
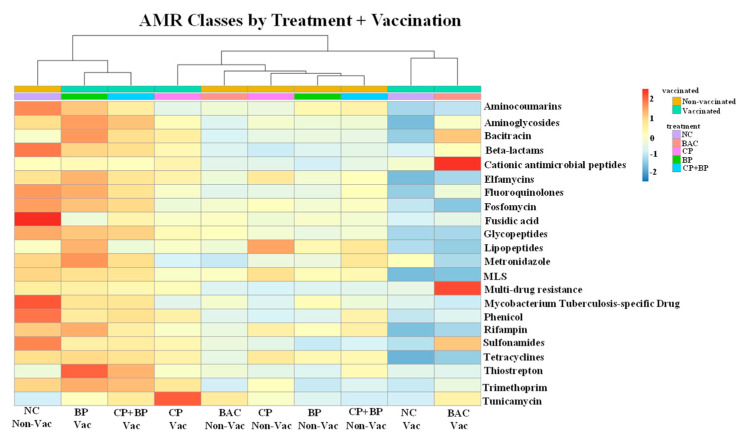
Heatmaps of relative abundances of antimicrobial resistance genes classes by treatment (NC, negative control; BAC, 55 ppm bacitracin; CP, 1% cranberry; BP, 1% blueberry; and CP + BP, a combination of 1% cranberry and 1% blueberry) and vaccination groups (Non-Vac = non-vaccinated; Vac = vaccinated). Vaccination effects were observed for gene classes conferring resistance to bacitracin (*p* = 0.02), cationic antimicrobial peptides (*p* = 0.001), multi-drug (*p* = 0.01), and trimethoprim (*p* = 0.04) by Kruskal–Wallis tests.

**Figure 4 microorganisms-11-01184-f004:**
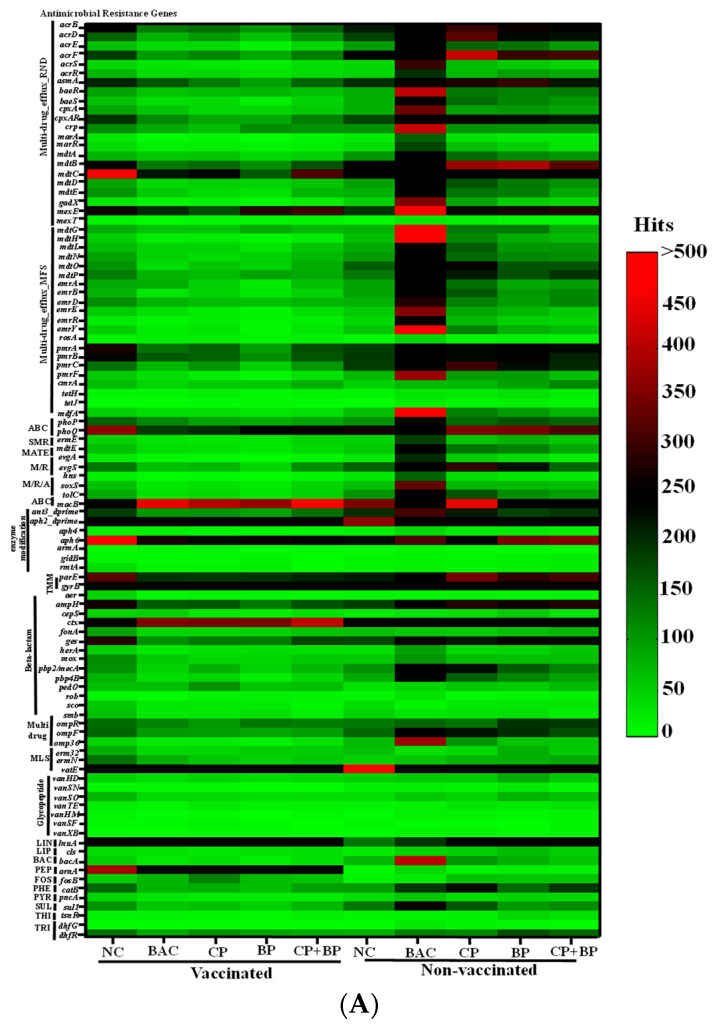
Heatmap showing changes in abundances of antimicrobial resistance genes identified in ceca from birds of studied dietary treatment groups and/or vaccination. (**A**) Shotgun metagenomics analysis results using AMR++: ABC, multi-drug_efflux_ABC; SMR, multi-drug_efflux_SMR; MATE, multi-drug_efflux_MATE; M/R, multi-drug_efflux_MFS/RND; M/R/A, multi-drug_efflux_ MFS/RND/ABC; TMM, target-medicated mutation; MLS, macrolides, lincosamides, streptogramins; LIN, lincosamides; LIP, lipopeptides; Bac, bacitracin; FOS, fosfomycin; PHE, phenicol; PYR, pyrazinamide; SUL, sulfonamide; TET, tetracycline; THI, thiostrepton; and TRI, trimethoprim. (**B**) Targeted-resistome sequencing results showing presence and absence of specific ARGs. Vac, vaccinated group; Non-Vac, non-vaccinated group; NC, basal diet as negative control; BAC, 55 ppm bacitracin; CP, 1% cranberry; BP, 1% blueberry; and CP + BP, a combination of 1% cranberry and 1% blueberry.

**Figure 5 microorganisms-11-01184-f005:**
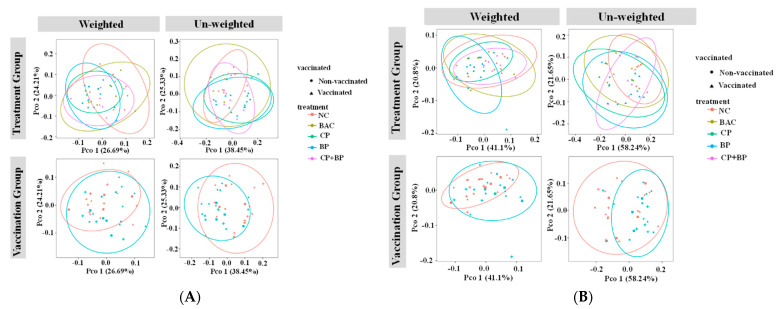
(**A**) Beta-diversity of total resistance genes (resistome) from bait capture. PCoA plot of weighted and unweighted Bray–Curtis showing clustering based on treatment (**Top**) and vaccination (**Bottom**). Significant dissimilarity was detected between treatment groups (weighted only *p* = 0.017). Significant resistome dissimilarity was detected between vaccination groups (weighted *p* = 0.012; unweighted *p* = 0.009). Adonis statistics suggest vaccination only explains a small proportion of the observed dissimilarity. NC, basal diet as negative control; BAC, 55 ppm bacitracin; CP, 1% cranberry; BP, 1% blueberry; and CP + BP, a combination of 1% cranberry and 1% blueberry. (**B**) Beta-diversity of aminoglycoside resistance genes from bait capture. PCoA plot of weighted and unweighted Bray–Curtis showing clustering based on treatment (**Top**) and vaccination (**Bottom**). No difference was detected among treatments (*p* = 0.122), but significant dissimilarity was detected between vaccination groups (weighted *p* = 0.011; unweighted *p* = 0.008). Adonis statistics suggest vaccination only explains a small proportion of the observed dissimilarity. NC, basal diet as negative control; BAC, 55 ppm bacitracin; CP, 1% cranberry; BP, 1% blueberry; and CP + BP, a combination of 1% cranberry and 1% blueberry.

**Figure 6 microorganisms-11-01184-f006:**
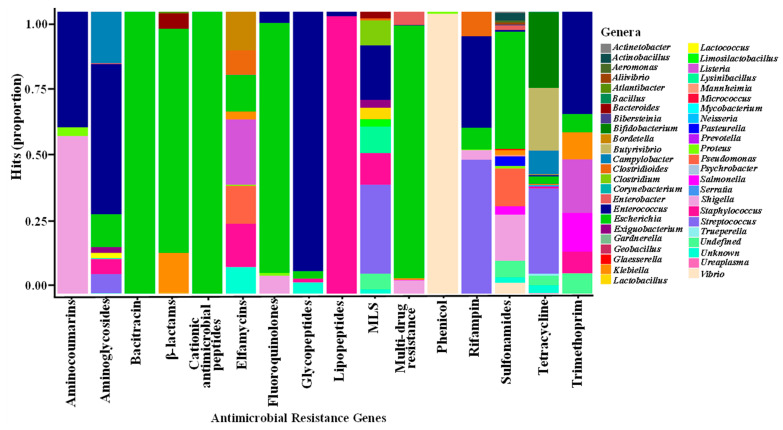
Spearman non-parametric correlations between antimicrobial resistance categories and microbiota genera of 21-day-old broiler chickens and ceca microbial genera (a minimum of 70% coverage). MLS, macrolides, lincosamides, and streptogramins.

**Figure 7 microorganisms-11-01184-f007:**
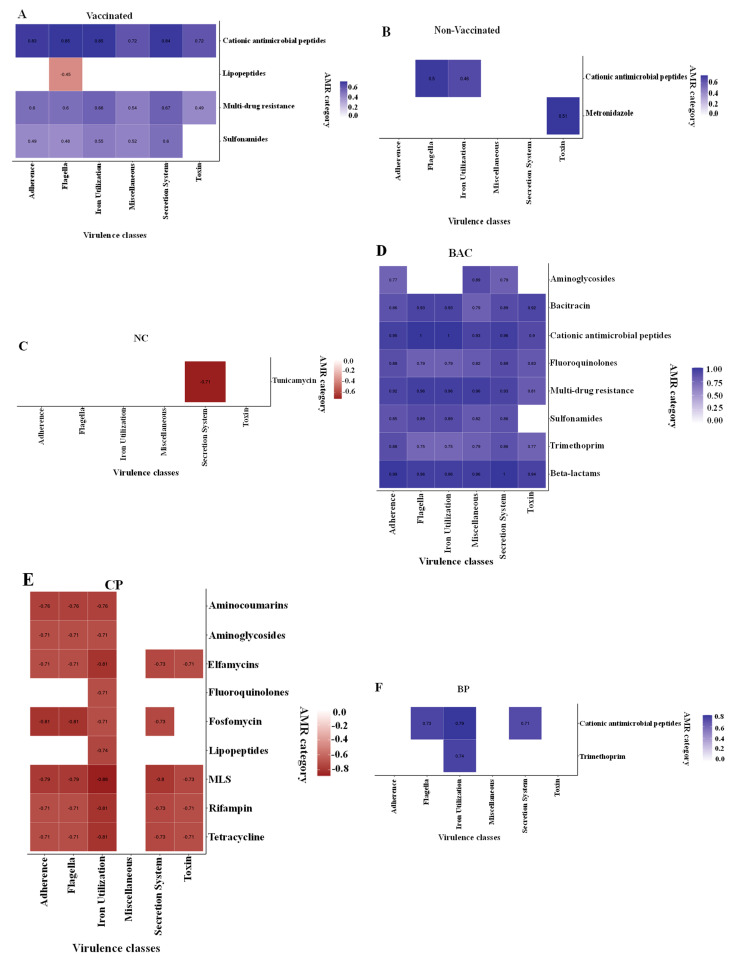
Spearman non-parametric correlations between VGs and antimicrobial resistance genes of 21-day-old broiler chickens and metabolic pathways by vaccination (**A**,**B**) and dietary treatment (**C**–**G**) groups. NC, basal diet as negative control; BAC, 55 ppm bacitracin; CP, 1% cranberry; BP, 1% blueberry; and CP + BP, a combination of 1% cranberry and 1% blueberry. The scale colors indicate whether the correlation is positive (closer to 1) or negative (closer to -0.5 or -1.0) between VGs and antimicrobial resistance genes. All correlations presented were statically significant (*p* < 0.05).

**Figure 8 microorganisms-11-01184-f008:**
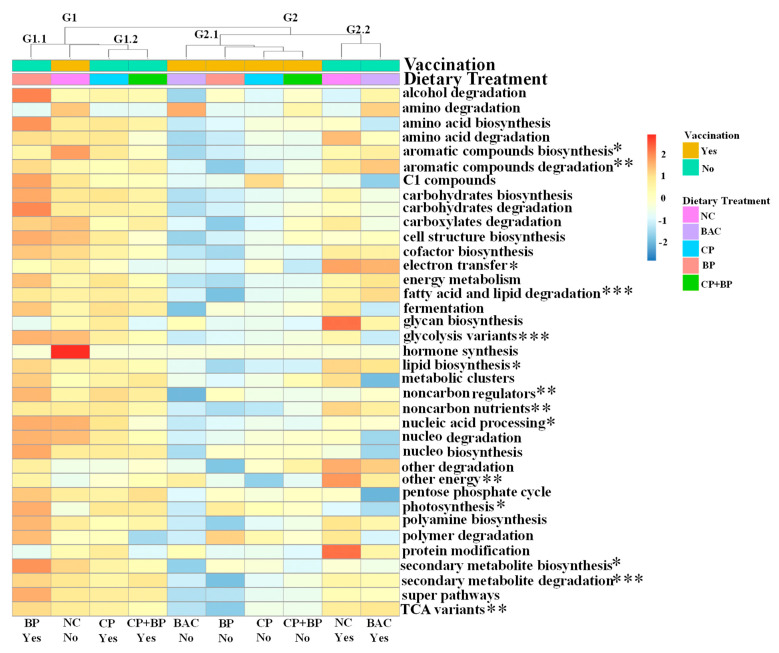
Heatmap showing changes in cecal metabolic pathways in studied dietary treatment groups. NC, basal diet as negative control; BAC, 55 ppm bacitracin; CP, 1% cranberry; BP, 1% blueberry; and CP + BP, a combination of 1% cranberry and 1% blueberry. Significantly affected by dietary treatment and/or vaccination: * 0.5 < *p* < 0.1 ((tendance); ** *p*< 0.05; and *** *p* < 0.01).

**Figure 9 microorganisms-11-01184-f009:**
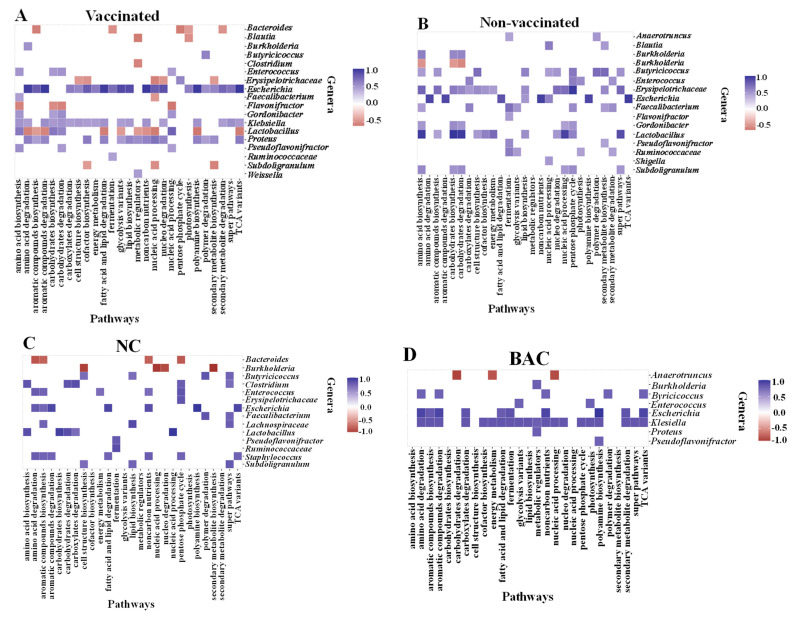
Spearman non-parametric correlations between ceca bacterial genera of 21-day-old broiler chickens and metabolic pathways by vaccination (**A**,**B**) and dietary treatment (**C**–**G**) groups. NC, basal diet as negative control; BAC, 55 ppm bacitracin; CP, 1% cranberry; BP, 1% blueberry; and CP + BP, a combination of 1% cranberry and 1% blueberry. The scale colors indicate whether the correlation is positive (closer to 1) or negative (closer to −0.5 or −1.0) between microbiota (genera) and metabolic pathways. All correlations presented were statically significant (*p* < 0.05).

**Table 1 microorganisms-11-01184-t001:** Relative abundances (%) of cecal microbiota at the levels of kingdom, phylum, family, genus, and species significantly affected by dietary treatment and/or vaccination.

Taxonomy	Non-Vaccinated ^1^					Vaccinated					SEM ^2^	Effects ^3^		
	NC	BAC	CP	BP	CP + BP	NC	BAC	CP	BP	CP + BP		Vac	Trt	Vac × Trt
*k_Bacteria*	81.60	82.71	93.95	94.30	89.76	63.25	86.34	85.71	89.03	92.79	2.301	ns	**	ns
*k_Eukaryotes*	0	0	0	0	0	0.1	0.02	0.07	0.01	0.01	0.011	*	ns	ns
*k_Virus*	18.40	17.29	6.05	5.70	10.24	36.65	13.64	14.22	10.96	7.21	2.297	ns	**	ns
*p_Firmicutes*	78.03	86.82	83.53	91.06	91.19	53.94	41.1	81.21	76.49	92.51	3.616	***	**	ns
*p_Proteobacteria*	1.82	0.65	0.39	0.15	0.56	3.71	14.15	4.68	6.30	1.57	1.106	**	ns	ns
*p_Viruses*	18.40	17.29	6.05	5.70	10.24	36.65	13.64	14.22	10.96	7.21	3.156	***	**	ns
*f_Lactobacillaceae*	43.91	63.80	81.29	68.29	62.14	31.67	37.37	52.67	58.38	68.04	3.646	ns	**	ns
*f_Oscillospiraceae*	13.08	6.01	3.94	5.07	6.52	7.67	8.42	6.48	8.69	6.48	0.621	ns	*	ns
*f_Enterobacteriaceae*	1.60	0.31	0.35	0.09	0.58	3.63	17.82	4.85	3.47	1.86	1.105	***	ns	*
*f_Eimeriidae*	0	0	0	0	0	0.1	0.02	0.07	0.01	0.01	0.011	*	ns	ns
*f_Retroviridae*	18.40	17.29	6.05	5.70	9.01	36.65	13.56	14.22	10.81	7.21	2.300	ns	**	ns
*g_Lactobacillus*	43.91	63.8	81.29	68.29	62.14	31.67	37.37	52.67	58.38	68.04	3.646	**	**	ns
*g_Butyricicoccus*	6.59	1.13	0.47	0.46	0.91	0.45	7.42	0.55	4.4	1.43	0.692	ns	ns	*
*g_Escherichia*	1.58	0.3	0.35	0.09	0.58	3.6	17.14	4.82	3.41	1.84	1.059	***	ns	*
*g_Eimeria*	0	0	0	0	0	0.1	0.02	0.07	0.01	0.01	0.011	*	ns	ns
*g_Retroviridae*	18.40	17.29	6.05	5.70	9.01	36.65	13.56	14.22	10.81	7.21	2.300	ns	**	ns

^1^ NC, basal diet; BAC, 55 ppm bacitracin; CP, 1% cranberry; BP, 1% blueberry; and CP + BP, a combination of 1% cranberry and blueberry. ^2^ SEM, standard error of the means. ^3^ Vac, main effects of vaccination; Trt, main effects of treatments; and Vac × Trt, interaction between vaccination and treatments. * Asterisks indicate significant statistical differences (one asterisk means a tendency level between 0.05 and 0.10; two asterisks mean a significance level less than 0.05; and three asterisks mean a significance less than 0.01); ns, not significant.

## Data Availability

The sequence read data of studied ceca bacterial metagenomes have been submitted to the Sequence Read Archive (SRA) database of the National Center for Biotechnology Information as FASTQ files under study accession number PRJNA666163.

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
