# Peer review of "Microbiome of Ceca from Broiler Chicken Vaccinated or Not against Coccidiosis and Fed Berry Pomaces"

_microorganisms, 2023, doi:10.3390/microorganisms11051184_

Round 1

Reviewer 1 Report

There is a bit of a disconnection between feed types and cocci vaccination. The two topics seem to be addressed separately throughout the entire paper. An important question here is: is the berry supplementation aiding the cocci vaccine in any way? If the answer is no, perhaps this paper should be divided into two: (1) the effects of cocci vaccination on the cecal microbiota of chickens, and (2) the effects of berry pomaces on the cecal microbiota of chickens. The main reason I make this suggestion is because this is a really heavy read. There is a lot of data presented here, and it becomes difficult to sort which results are relevant or not considering the 10 groups altogether (5 feed types given to birds vaccinated or unvaccinated). The authors do a decent job going over the results with the reader. However, there are 14 pages of results, including 9 multipart figures, but less than 3 pages of discussion. The paper could be more objective and focus their results on what they actually want to show/discuss instead of making the reader confused with a lot of information that the authors didn’t seem relevant enough to discuss in detail.

Some conclusions are a bit of a stretch. For example, the sentence “our results provided new insights on the potential impacts of live oocyst vaccines on bacterial virulence genes in broiler gut” (lines 546-548) is not fully supported by their results. Similarly, there is no supporting evidence that “AMR and its related genes in ceca could be promoted by coccidiosis vaccination” (lines 568-569). These observations might suggest a correlation but could also have been a random or false association that does not indicate causation whatsoever. A less complex and more straight-forward experimental design would be needed to address these assumptions about the effects of the cocci vaccine in the gut microbiome.

In summary, the data from this paper is a good addition to the field and there isn’t anything technically wrong with it that I can notice. My suggestion is to make it more objective, using less jargon and providing more concise results that still support their conclusions. Some of the conclusions or speculations could be softened to satisfy the “correlation vs causation” matter. Even if the authors do not accept my suggestions, this manuscript is still suitable for publication as is.

Author Response

Section (A): Response to the Suggestions and Comments by the Reviewer 1

R1 Comment #1. An important question here is: is the berry supplementation aiding the cocci vaccine in any way? If the answer is no, perhaps this paper should be divided into two: (1) the effects of cocci vaccination on the cecal microbiota of chickens, and (2) the effects of berry pomaces on the cecal microbiota of chickens. There are 14 pages of results, including 9 multipart figures, but less than 3 pages of discussion.

Authors’ response: Thanks. Improved growth performance parameters are vital for broiler industries. Intestinal diseases (such as coccidiosis) can significantly impair these parameters. Thus, efficient intestinal diseases control strategies are necessary. Coccidiosis is induced by Eimeria, spp. The efficacy of coccidiosis vaccines to prevent incidence of coccidiosis has been subject of several published studies by other scientists. But coccidiosis vaccine may compromise growth performance of broiler chickens. As stated in the introduction, our previous studies have documented the positive effect of berry pomaces and extracts feed supplementation on lowering Eimeria oocyst numbers and prevalence of intestinal lesions due to coccidiosis. The main objective of the present study was to analyze cecal microbiome (microbiota, virulome, resistome and metabolites) in cranberry and blueberry pomaces-fed broiler chickens vaccinated or not vaccinated against coccidiosis. Our data indicate that berry pomace feeding could help to improve vaccine effects. In this case, it has more rational to keep both vaccination and berry pomaces together in one paper.

We agreed that there are too many contents of results. These data are all important because they include cecal microbiota (Table 1), virulence genes categories (Figure 1), virulence genes (Figure 2), antimicrobial resistance genes classes (Figure 3), antimicrobial resistance genes (figure 4), Beta-diversity of resistance genes by Capture (Figure 5), correlation between antimicrobial resistance categories and cecal microbiota genera (Figure 6), correlation between antimicrobial resistance genes and virulence genes (Figure 7), metabolic pathways (Figure 8), and correlation between cecal microbiota genera and metabolic pathways (Figure 9). Especially the correlation analyses are important to explain the possible mechanisms of berry pomaces on microbiome. The discussion is short since most of the mechanisms are still unknow which deserves our further study.

R1 Comment #2. Some conclusions are a bit of a stretch. For example, the sentence “our results provided new insights on the potential impacts of live oocyst vaccines on bacterial virulence genes in broiler gut” (lines 546-548) is not fully supported by their results. Similarly, there is no supporting evidence that “AMR and its related genes in ceca could be promoted by coccidiosis vaccination” (lines 568-569). These observations might suggest a correlation but could also have been a random or false association that does not indicate causation whatsoever. A less complex and more straight-forward experimental design would be needed to address these assumptions about the effects of the cocci vaccine in the gut microbiome.

Authors’ response: Thanks. Lines 546-548: in this study, the vaccinated group had a higher overall prevalence of virulence genes compared to the non-vaccinated group. The coccidiosis are live oocytes. Since no previous studies have ever reported, our study could be a new insight of impacts of the live oocytes on bacterial virulent genes. Since reasons are still unknown however, we added “, further studies are warranted to reveal the mechanisms”.

Lines 568-569: this conclusion was made since we found a higher prevalence of antimicrobial resistance genes and categories in vaccinated group compared to non-vaccinated group. Since the mechanism is still unknown, we added “Further study is required to understand the mechanisms of the impacts of coccidiosis vaccines on AMR.”

Reviewer 2 Report

This article analyses concerns cecal microbiota, virulome, resistome and metabolites in American cranberry and lowbush blueberry pomaces-fed broiler chickens vaccinated or not vaccinated against coccidiosis

Introduction - overall is complete, line 72-74 does not make much sense in the way that is presented. 

methods - the scientific names of the berry where it would be interesting. A small summary of the technique would be more informative in the berry preparation section. line 102 previously describes where?  

results: well presented. Figure 4 is very small and is difficult to understand

discussion: could be improved

conclusion: maybe separate from discussion, it is a little confusing as it is

Author Response

Section (B): Response to the Suggestions and Comments by the Reviewer 2

R2 Comment #1. Introduction - overall is complete, line 72-74 does not make much sense in the way that is presented. 

Authors’ response: Thanks. The sentence has been changed to “Metagenomics Phylogenetic Analysis (MetaPhlAn) [17], Short Reading Sequencing Typing (SRST2) [18], AmrPlusPlus (AMR++) [19] and HUMAnN (v2.0) are four tools to analyze metagenomics data, which have been frequently used to investigate microbiota taxonomy, virulome, resistome, and metabolic pathways, receptively [21]”.

R2 Comment #2. Methods - the scientific names of the berry where it would be interesting. A small summary of the technique would be more informative in the berry preparation section. line 102 previously describes where? 

Authors’ response:  The scientific names of the berry were added: cranberry (Vaccinium macrocarpon) and blueberry (Vaccinium angustifolium).

The summary of the previously published pomace preparation methods has been also added “Briefly, frozen A hydraulic rack and cloth were used to press berry fruits at 1000 to 3000 psi for juice removal. The pomaces then freeze-dried and grounded through a 2-mm mesh screen using a cutting mill (SM 2000 Retsch, Haan, Germany)”.

The “previously described” means that the protocol of Genomic DNA extracted using the QIAamp DNA Stool Mini Kit (Qiagen) was previously described in Das et al. (2021). In general, to increase the DNA quality and minimize biases to ensure reproductivity, bead-beating method was used to enhance the yield of DNA from Gram-positive bacteria and denaturants including guanidine isothyocynate and β-mercaptoethanol to shield DNA from nucleases after cell lysis. We added the followed statement “Briefly, to increase the DNA quality and minimize biases to ensure reproductivity, bead-beating method and denaturants including guanidine isothyocynate and β-mercaptoethanol were used” in the manuscript.  

R2 Comment #3. Results: well, presented. Figure 4 is very small and is difficult to understand.

Authors’ response: Thanks. We did our best to improve the quality of the of Figure 4.

R2 Comment #4. Discussion: could be improved. Conclusion: maybe separate from discussion, it is a little confusing as it is

Authors’ response: Thanks. The “Discussion” was slightly modified for clarification, and the “Conclusion” has been separated from the “Discussion.”